# Role of the Nrf2 Signaling Pathway in Ovarian Aging: Potential Mechanism and Protective Strategies

**DOI:** 10.3390/ijms241713327

**Published:** 2023-08-28

**Authors:** Xiaofan Gao, Bo Wang, Yibao Huang, Meng Wu, Yuting Li, Yinuo Li, Xiaoran Zhu, Mingfu Wu

**Affiliations:** 1National Clinical Research Center for Obstetrical and Gynecological Diseases, Department of Obstetrics and Gynecology, Tongji Hospital, Tongji Medical College, Huazhong University of Science and Technology, Wuhan 430030, China; gaoxiaofan1998@163.com (X.G.); bowangtjhp@126.com (B.W.); huangyibaogxmu@163.com (Y.H.); mengwu@tjh.tjmu.edu.cn (M.W.); lytingdoc@163.com (Y.L.); lynuo96699@163.com (Y.L.); kara232009@hotmail.com (X.Z.); 2Key Laboratory of Cancer Invasion and Metastasis, Ministry of Education, Wuhan 430030, China

**Keywords:** ovarian aging, oxidative stress, Nrf2, antioxidant therapy

## Abstract

The ovary holds a significant role as a reproductive endocrine organ in women, and its aging process bears implications such as menopause, decreased fertility, and long-term health risks including osteoporosis, cardiovascular disorders, and cognitive decline. The phenomenon of oxidative stress is tightly linked to the aging metabolic processes. More and more studies have demonstrated that oxidative stress impacts both physiologic and pathologic ovarian aging, and the nuclear factor erythroid 2-related factor 2 (Nrf2) pathway plays a crucial role in regulating the antioxidant response. Furthermore, various therapeutic approaches have been identified to ameliorate ovarian aging by modulating the Nrf2 pathway. This review summarizes the important role of the Nrf2/ Kelch-like ECH-associated protein 1 (Keap1) signaling pathway in regulating oxidative stress and influencing ovarian aging. Additionally, it highlights the therapeutic strategies aimed at targeting the Nrf2/Keap1 pathway.

## 1. Introduction

With advancing modernization and the increasing human longevity, the disease burden of aging should not be underestimated, and aging-related diseases have a serious impact on the quality of life [1,2]. Unlike other organs, which can undergo tissue regeneration, the ovary has a limited pool of ovarian follicles at birth. As these follicles are used up over time, there is no significant regenerative capacity for the ovary to counteract the effects of aging. Studies have found that reproductive aging occurs prior to somatic aging [3]. Additionally, the increased global average life expectancy means an increased menopausal status over a woman’s lifetime. Therefore, ovarian aging is even more important to a woman’s quality of life.

Ovarian aging is the ongoing process of decline and the eventual depletion of ovarian function [4]. It not only leads to a reduced reproductive capacity, fertility, and menopausal syndromes, but also increases the risk of aging-related diseases such as osteoporosis, cognitive impairment, cardiovascular disease, and cancer [5,6,7,8]. It has been shown that all-cause mortality and cancer rates are higher in the premature ovarian failure (POF) population than in the normal menopausal age group [9]. Studies suggest that ovarian aging is the pacemaker for other organs aging in women and is crucial to a woman’s healthy lifespan. On the one hand, the function of the ovary, including the ovarian reserve and oocyte quality, declines with age. On the other hand, several factors can damage ovarian function to different degrees, leading to pathological ovarian aging [10]. There are various molecular mechanisms involved in ovarian aging, such as epigenetic molecular alterations, autophagy, apoptosis, inflammation, and oxidative stress [11]. Notably, oxidative stress is one of the most classical theories of aging and plays an indispensable role in the development of ovarian aging.

Numerous research have demonstrated that the ovarian microenvironment’s oxidative stress condition can lead to pathological harm. The harm includes the interruption of meiosis in oocytes, apoptosis of granulosa cells, and dysfunction of the corpus luteum. Consequently, this expedites the progression of ovarian aging [12]. Oxidative stress is a series of adaptive responses caused via an imbalance between reactive oxygen species (ROS) and antioxidants when the cell is exposed to various stressors [13,14]. It is the most common stress event and affects various organs of the body, such as the heart, lungs, nervous system, reproductive system, and so on [15,16,17]. Meanwhile, oxidative stress is an important theme related to aging. Aging is often accompanied by an increase in oxidative stress, accumulation of ROS, leads to suspension of normal cellular functions. Meanwhile, cells modulate the production of anti-oxidant enzymes to remove ROS for self-saving. Nuclear factor erythroid 2-related factor 2 (Nrf2) is an important transcription factor of the oxidative stress response. It is regulated via Kelch-like ECH-associated protein 1 (Keap1) [18]. Nrf2 can promote the expression of downstream antioxidant enzymes, such as heme oxygenase 1 (HO-1), NADP(H) quinone dehydrogenase 1 (NQO1), and glutathione-S-transferases (GSTs). This mechanism effectively mitigates intracellular oxidative stress and minimizes cellular harm induced by reactive oxygen species. As an important regulator of oxidative stress, the Nrf2/Keap1 pathway also plays an important role in aging and various diseases, including preeclampsia, traumatic brain injury, and cancer [19,20,21,22,23]. In aging and some diseases, the Nrf2/Keap1 pathway is a protective factor, but may also act as a double agent in cancer [24]. For example, in Hutchinson–Gilford Progeria Syndrome, Nrf2 transcriptional activity and downstream antioxidant products reduce. The activation of Nrf2 has the potential to reverse the observed phenotype. Nrf2-mediated antioxidant responses were confirmed to be a key factor in the premature aging phenotype [25]. Nrf2 has been proved to be related to regulating the hallmarks of aging. In this review, we take Nrf2 as a starting point to explore the molecular mechanisms of the oxidative stress in ovarian aging and summarize some therapeutic means targeting Nrf2 to intervene in ovarian aging.

## 2. Nrf2/Keap1 Pathway

Nrf2 is the member of the Cap’n’Collar (CNC)-basic leucine zipper (bZIP) family of transcription factors, consisting of seven domains (Figure 1A) [26]. The bZIP motif in the Neh1 domain plays a crucial role in facilitating the interaction between Nrf2 and the antioxidant response element (ARE) [27]. The Neh2 domain is the most important regulatory domain in regulating the interaction between Nrf2 and Keap1 via the Glu-Thr-Gly-Glu (ETGE) and Asp-Leu-Gly (DLG) motifs [28]. The Neh3, Neh4, and Neh5 domains are responsible for the transcriptional activation of Nrf2 [29,30]. The Neh6 domain has been identified as an indispensable role in the Keap1-independent degradation of Nrf2 and regulates the stability of Nrf2 [31].

Keap1 contains five domains (Figure 1B) [32], including the N-terminal region (NTR), the broad-complex, tramtrack, and bric-a-brac domain (BTB), the intervening region (IVR), the double glycine repeat domain (DGR), and the C-terminal region (CTR). The DGR domain, alternatively referred to as the Kelch domain, serves as the binding site for Keap1 to interact with the Neh2 segment of Nrf2 [33].

Nrf2/Keap1/ARE is the predominant signaling pathway in the oxidative stress response (Figure 1C). In a state of homeostasis, the transcription factor Nrf2 forms a complex with the protein Keap1 and undergoes ubiquitination, leading to its later degradation within the cytosol [34]. Electrophiles effectively suppress the activity of the Keap1 complex and facilitate the generation of free Nrf2 in the presence of oxidative stress. Nrf2, upon liberation, migrates to the nucleus and establishes a heterodimeric association with small musculoaponeurotic fibrosarcoma (sMaf). Subsequently, this heterodimeric complex undergoes aggregation and binds to ARE. The binding of Nrf2 to the ARE leads to transcriptional processes and facilitates the activation of downstream antioxidant genes [35], including HO-1, NQO1, and GSTs [36,37].

## 3. Role of the Nrf2 Signaling Pathway in Ovarian Aging 

Nrf2, an important regulatory molecule in oxidative stress, plays an important role in ovarian aging. Many studies have confirmed that Nrf2 acts as a protective factor against physiological and pathological ovarian aging and as a cellular defense against the effects of aging (Figure 2) [12,38,39,40]. In the following section, the significant role of Nrf2 will be discussed in relation to four key aspects: age-related ovarian aging, environmental pollutant-related ovarian aging, unhealthy lifestyle-related ovarian aging, and chemotherapy-related ovarian aging.

### 3.1. Nrf2 Signaling and Age-Related Ovarian Aging 

Generally speaking, after the age of 35, women’s ovarian function begins to decline. At the age of 49–52, women enter menopause, when women’s ovarian function completely fails [41]. As individuals age, there is a notable increase in oxidative stress levels and a corresponding decrease in ovarian antioxidant capacity. The presence of an intracellular redox imbalance has been found to have an impact on both the oocytes and granulosa cells (GCs) [42,43,44]. Nrf2 is an important regulator of cellular antioxidant responses, and its expression showed a decreasing trend with increasing age in ovarian tissues [45,46,47]. Sindan et al. found that the Nrf2 protein was mainly localized in the granulosa cells and oocytes of the secondary and antral follicles in mouse ovaries [45]. In addition, *Nrf2^−/−^* mice had fewer remaining primordial follicles at 10–12 months of age than wild mice [38]. This indicated that the antioxidant capacity of the ovary might decrease with age and that *Nrf2* knockout accelerated ovarian aging.

Nrf2 and Keap1 proteins are predominantly expressed in the cytosol of granulosa cells [48]. When granulosa cells are exposed to oxidative stress, ROS will increase and mitochondrial activity will reduce. In this condition, Nrf2 is activated and increases the expression of downstream antioxidant products, such as peroxiredoxin 1 (PRDX1), superoxide dismutase1 (SOD1), catalase (CAT), HO-1, thioredoxin 1 (TXN1), and NQO1 [39,48]. Meanwhile, granulosa cells release large amounts of exosomes which contain *Nrf2*-encoded mRNA and antioxidant molecules. The exosomes are taken up by neighboring cells to jointly protect against oxidative damage [39]. Inflammatory aging (inflamm-aging) is a persistent and mild inflammatory condition that develops as age increases. In recent years, research has indicated that the process of inflammatory aging is a significant factor in the development of premature ovarian insufficiency (POI) [49]. The aberrant overexpression of transferrin and ferritin in aging rat ovaries had been found to contribute to inflamm-aging via the nuclear factor kappa-light chain-enhancer of activated B lymphocytes (NF-κB) /inducible nitric oxide synthase (iNOS) pathway. This, in turn, downregulated the Nrf2/glutathione peroxidase 4 (GPX4) signaling pathway, leading to decreased antioxidant defenses and increased oxidative stress in ovarian granulosa cells [46].

Age-related decline in oocyte quality and quantity leads to progressive fertility and eventual natural infertility [50]. The expression of Nrf2 in oocytes also decreases with advancing age, and the depletion of Nrf2 has detrimental effects on oocyte maturation and spindle organization. These effects are mediated via the sirtuin 1 (SIRT1)/Nrf2/Cyclin B1 signaling pathway. Conversely, the overexpression of Nrf2 alleviated maternal age-related meiotic defects [40].

In summary, the Nrf2 expression level declines with age in both oocytes and granulosa cells and affects cellular function by influencing cellular oxidative stress and apoptosis, etc. (Figure 3). Furthermore, in the presence of oxidative stress, granulosa cells release exosomes that contain *Nrf2*-encoded mRNAs. These exosomes are then internalized via the surrounding granulosa cells to collectively defend against oxidative stress. However, it remains unclear whether there is intercellular communication between granulosa cells and oocytes in response to increased oxidative stress with increasing age. Further investigation is required to examine the potential association between granulosa cells and oocytes in the aging ovary via the Nrf2 signaling pathway.

### 3.2. Environmental Pollutant-Related Ovarian Aging and Its Association with Nrf2 Signaling 

With modern industrial development, there is an increase in environmental pollutants. Consequently, there has been a progressive rise in human susceptibility to environmental pollutants. Many studies have demonstrated that a wide range of environmental pollutants have detrimental effects on multiple organ systems, such as the liver, kidney, and nervous system [51,52]. In women, they even reduce the ovarian reserve function and mediate the onset of ovarian aging [53,54,55,56]. This article provides an overview of four primary classifications of pollutant-induced ovarian aging, namely endocrine-disrupting chemicals (EDCs), industrial compounds, heavy metals, and nanoparticles.

#### 3.2.1. Endocrine-Disrupting Chemicals 

Endocrine-disrupting chemicals are a class of environmental pollutants that have been shown to have adverse effects on human reproductive function. EDCs cause abnormalities in the normal homeostasis or reproduction by interfering with hormone biosynthesis, metabolism, or behavior [57]. For instance, Bisphenol A (BPA) is widely used in the production of various plastics, such as electronics, paper products, water pipes, plastic bottles, food containers, etc. [58]. A long-term exposure (14 weeks) to BPA was found to cause the degeneration of ovarian follicles. This degeneration was characterized by a significant decrease in the number of primary and developing follicles, as well as an increase in the number of atretic follicles. These effects were attributed to the downregulation of the *Nrf2* gene expression [59]. Bisphenol S (BPS) is often regarded as a potentially less harmful substitute for BPA. However, research has shown that BPS had detrimental effects on ovarian function and reduced the expression of Nrf2, similar to the effects observed with BPA. What is more, these were rescued via melatonin via the modulation of the SIRT-1/Nrf2/NF-κB pathway [60].

#### 3.2.2. Industrial Compounds

In addition to EDCs, there are many occupationally exposed industrial compounds that can influence ovarian reserve and function, leading to the development of POF. 4-Vinylcyclohexene diepoxide (VCD) is generated from the precursor compound 4-Vinylcyclohexene via cytochrome P450-catalyzed epoxidation. This chemical compound has been identified as a potential occupational ovarian toxicant. VCD exposure for 15 days decreased the primary follicle number and destroyed the structure of small follicles in mice, inducing the early depletion of functional follicles. Whereas *Nrf2^−/−^* mice showed more severe follicle loss and POF compared to wild-type mice exposed to VCD. Wild-type mice showed increased ROS levels and an activated Nrf2-dependent anti-ROS protective response when exposed to VCD. However, *Nrf2^−/−^* inhibited this antioxidant response and exacerbated the ovarian toxicity of VCD [61]. 1-Bromopropane, another ovarian toxicant, induced disruptions in the estrous cycle of rats. Additionally, there was a notable decrease in the number of sinus follicles and growing follicles [62]. In vitro, 1-Bromopropane exposure increased ROS levels and decreased the expression of important antioxidant molecules such as Nrf2, HO-1, etc. Consequently, the increased ROS levels triggered apoptosis in ovarian cells [63].

#### 3.2.3. Heavy Metals

Heavy metal pollution can also have a serious impact on reproductive function. Cadmium, for example, is a heavy metal that is highly toxic to humans, animals, and plants, even at low doses [64]. A study suggested that Cadmium (Cd) exposure caused premature ovarian failure characteristics [65] via increasing oxidative stress and inflammation in the ovaries. In the present study, the levels of ROS and malondialdehyde (MDA) in fish ovaries exposed to acute cadmium (Cd) were not significantly increased [66]. However, the expression levels of Nrf2, SOD1, and CAT in the ovaries were upregulated. Based on these findings, we hypothesized that the oxidative stress caused by acute Cd exposure was counteracted via the prompt activation of antioxidant responses [66]. In vitro, exposure to Cd decreased the expression of Nrf2 and the antioxidant genes CAT, SOD, and NQO1, resulting in the increased apoptosis of granulosa cells. The involved mechanism includes m6A methylation modification to regulate the activity of AKT serine/threonine kinase 1 (AKT) and Nrf2 [67]. Lead, which humans are widely exposed to on a daily basis, can cause impaired fertility and hormone disruption in women [68,69]. The short-term (7 days) exposure to lead resulted in elevated levels of reactive oxygen species (ROS) in the oocytes of mice. The increased ROS induces mitochondrial dysfunction and impairs oocytes maturation, embryo development, and fertilization. At the same time, they observed the activation of the intracellular Nrf2/Keap1 signaling pathway and increased antioxidant enzyme transcripts [70]. The ovarian function of laying hens was found to be impaired as a result of long-term exposure (9 weeks) to lead or mercury. This exposure led to an increased rate of ovarian follicle atresia. Additionally, there was an elevation in MDA and Keap1 expression levels, and a decrease in the expression of Nrf2 and antioxidant enzymes. These alterations in gene expression patterns further contributed to the impairment of ovarian function [71,72]. Other regulatory mechanisms included the Nrf2/NF-κB signaling pathway [73].

#### 3.2.4. Nanoparticles

Nanoparticles are widely applied in many fields due to their excellent biocompatibility, corrosion protection, and so on. As nanoparticles become more prevalent in the environment, the likelihood of exposure to the nanoparticles increases. Recently, nanoparticles have been found to cause harmful effects when ingested by humans or animals. They cause damage to the digestive, respiratory, and nervous systems [74,75,76,77,78]. Therefore, more attention should be given to the biosecurity of nanoparticles. Significantly decreased serum estradiol and progesterone levels, impaired follicular structure, and increased number of atretic follicles were found in mice that were exposed to Zinc oxide nanoparticles (ZnO NPs) for 7 days. Meanwhile, the expression levels of antioxidant-related genes, endoplasmic reticulum (ER) stress-related genes, and apoptosis-related genes were upregulated, but SOD activity was downregulated. The toxic effects of ZnO NPs on the ovaries of female mice may be related to the destruction of the antioxidant enzyme system [79]. Nano-PS exposure for 2 weeks damaged mouse ovaries and reduced the serum estrogen (E_2_), anti-Mullerian hormone (AMH), and fertility. In vitro, Nrf2 levels in cells treated with Nano-PS were significantly higher than the control at 24 h, but lower after 48 h. In addition, treatment with Nrf2 inhibitors further confirmed that Nrf2 exerted cytoprotective effects during exposure [80].

#### 3.2.5. Conclusions

In brief, the underlying mechanisms of reproductive toxicity caused by various environmental pollutants involve oxidative stress and apoptosis (Figure 4). However, the expression of the Nrf2/Keap1 pathway has shown variability across different experimental studies. Based on this observation, we propose a hypothesis that the Nrf2/Keap1 pathway is initially activated as a defense against oxidative damage during the early stages of exposure to environmental pollutants. However, if the antioxidant system fails to properly remove excess ROS within the cell, it may result in a decrease in antioxidant enzyme activity and suppression of the Nrf2/Keap1 pathway. In addition, heavy metals may directly reduce antioxidant enzyme activity to damage cells. Moreover, gene knockdown and inhibitors have confirmed that Nrf2 is a protective factor against the reproductive toxicity of various pollutants.

### 3.3. The Relation between the Keap1/Nrf2 Pathway and Unhealthy Lifestyle-Related Ovarian Aging

The percentage of women among the total smoking population is on rise. As the harmful effects of smoking are well established, it has major concerns for females due to its adverse effects on the reproductive system [81]. Cigarette smoke contains thousands of harmful components, many of which are toxic to the reproductive organs. The probability of early menopause is positively associated with the intensity, duration, cumulative dose, and earlier initiation of smoking [82]. In vitro experiments revealed that cigarette smoke extract (CSE) inhibited the proliferation of ovarian granulosa cells. The increase in Keap1 expression and the decrease in Nrf2 expression resulted in a decrease in cellular protection against oxidative stress [83]. For oocytes, cigarette smoke led to follicular depletion by increasing oxidative stress and inducing apoptosis in developing follicles. Microarray analysis indicated that Nrf2-mediated oxidative stress played a role in the toxic effects of smoking on the ovaries [84].

Above all, CSE induced oxidative stress via the Keap1/Nrf2 pathway in ovarian cells (Figure 5). More in-depth studies are needed to understand how CSE-mediated changes in the downstream and upstream signals of the Nrf2 pathway affect ovarian function. Further studies are necessary to verify whether the various components found in cigarettes have similar effects on the ovaries and to determine if there are other mechanisms at play in this process.

### 3.4. Chemotherapy-Related Ovarian Aging

Chemotherapeutic drugs are used to induce the death of cancer cells to treat cancer [85,86]. Due to the non-selective nature of chemotherapeutic drugs, chemotherapeutic drugs will also significantly damage normal cells except for tumor cells. Consequently, this indiscriminate action results in impaired organ function, such as neurological damage, liver, and kidney dysfunction, and reproductive system impairment [87,88,89,90]. Chemotherapeutic drugs induce oxidative stress and apoptosis in normal cells, leading to the impairment of organ function. Oxidative stress is a significant contributor to the damage inflicted on other organs via chemotherapeutic drugs. The protective function of Nrf2 has been established in other organs [91], and its role in safeguarding against chemotherapy-induced ovarian aging has also been highlighted.

Zhang et al. found significantly higher serum follicle stimulating hormone (FSH) and luteinizing hormone (LH) levels and a higher incidence of POF in chemotherapy patients [92]. The study showed that the occurrence of POF was unavoidable for female patients of reproductive age exposed to cyclophosphamide (CP). This condition was characterized by the irreversible cessation of menstruation and infertility [93,94]. Conventional chemotherapeutic drugs, such as cyclophosphamide (CTX), paclitaxel (Tax), doxorubicin (Dox), and cisplatin (Cis), reduced ovarian volume with ovarian fibrosis and decreased the ovarian reserve. The mechanism of injury involved increased ROS, the upregulation of Keap1, the downregulation of Nrf2, and increased apoptosis [92]. Mice exposed to DOX for 2 weeks increased mRNA expression levels of Nrf2, HO-1, and CAT. However, there was a decrease in the protein levels of Nrf2, and MDA increased as SOD decreased [95]. CP significantly inhibited the expression of Nrf2 and promoted the production of the inhibitory upstream factor Keap1. This resulted in a decrease in the downstream products, NQO-1 and HO-1 [96].

In chemotherapeutic drug-induced ovarian damage (Figure 6), the Nrf2 protein was decreased, but its RNA levels were elevated [95]. This indicates that cellular adaptive defense mechanisms are activated when cells or tissues are exposed to harmful substances. However, these defenses do not completely prevent the damage caused via chemotherapeutic drugs. Therefore, chemotherapeutic drugs subsequently lead to a decrease in the functionality of antioxidant enzymes and the inhibition of the Nrf2/Keap1 pathway.

## 4. Targeting Nrf2 Signaling as Therapeutic Interventions in Ovarian Aging 

Previous studies have confirmed the important role of oxidative stress triggered by Nrf2 in ovarian aging. So, how to delay or reverse ovarian aging by targeting Nrf2 has become an important research question. According to recent researches, there are various therapeutic approaches that can affect the process of ovarian aging by modulating Nrf2 activity. The therapeutic approaches include Traditional Chinese Medicine, nutrients or endogenous hormone supplements, natural plant and animal extracts, stem cell therapies, and other compounds.

### 4.1. Traditional Chinese Medicine

Electroacupuncture (EA) is a popular therapy for POI in China. EA activates the Keap1/Nrf2/HO-1 signaling pathway and rescues CTX-induced ovarian dysfunction, which includes a disturbed motility cycle, decreased AMH and E_2_ hormone levels, and an increased atretic follicle number [97].

Traditional Chinese medicinal formulations have been used for thousands of years in the treatment of gynecological disorders. The traditional Chinese medicine formula Bu Shen Huo Xue Tang (BSHXT) is a modified version of the traditional Qing’e Pill formula from the Song Dynasty’s “Taiping Huimin and Agent Bureau”. What is more, it is commonly employed in clinical settings for the management of POI. It increased the levels of E_2_ and AMH and decreased the levels of FSH and LH, upregulated the expression of Nrf2, HO-1, and NQO1 genes, and increased the levels of SOD in an autoimmune mouse POI model. Bu Shen Huo Xue Tang enhanced the antioxidant capacity to protect the ovary through activating the Nrf2/Keap1 signaling pathway [98]. Another traditional Chinese medicinal preparation Si-Wu-Tang, also known as “the first prescription of gynecology” in China. It was widely used as the basic formula in regulating the menstrual cycle and treating infertility. The Si-Wu-Tang treatment improved estrogen levels, follicle numbers, and produced better fertility outcomes in POF mice via the activation of the Nrf2/HO-1 and STAT3/HIF-1α/VEGF pathways [99].

### 4.2. Nutrients or Endogenous Hormone Supplements

Vitamin C is a water-soluble vitamin with strong reducing properties and can be used as a nutritional supplement and antioxidant. It participates in complex metabolic processes in the body, promotes growth, and strengthens resistance to disease. Vitamin E is a fat-soluble vitamin that is widely used as an excellent antioxidant and nutrient in clinical, food, nutraceutical, and cosmetic industries. Studies proved that vitamin C and vitamin E alleviated oxidative stress-induced apoptosis by activating the Nrf2 signaling pathway to decrease intracellular ROS levels and inhibit caspase activity [63,100]. Furthermore, *Nrf2* knockdown impaired the cytoprotective effects of vitamin E on granulosa cells [100].

Arginine is an amino acid previously reported to exert antioxidant effects by upregulating the expression of ARE-driven antioxidant products via the activation of the Nrf2/Keap1 pathway [101]. Arginine had been shown to ameliorate the prolongation of estrous cycle days, hormone level disorders in multiparous Hu sheep, and increase the antioxidant capacity of ovarian tissues via the Nrf2/Keap1 pathway [102,103].

N-acetylcysteine is a powerful antioxidant. By decreasing ROS levels and upregulating the antioxidant products Nrf2, GPX4, and HO-1, it attenuates apoptosis and ferroptosis in GCs, thereby reversing the decline in ovarian function induced by cisplatin [94].

Melatonin (MT) is one of the hormones secreted by the pineal gland of the brain, which has powerful neuroendocrine immunomodulatory activity and antioxidant capacity. It stimulated the expression of key redox/survival markers such as SIRT1 and Nrf2, and enhanced ovarian antioxidant capacity to ameliorate the decline in ovarian function caused by BPS [60].

### 4.3. Natural Plant and Animal Extracts

Recent evidence has demonstrated that many natural plant extracts mediate Nrf2 activation to retard ovarian aging. *Euterpe oleracea* enhanced blastocyst development and mitochondrial function in aged mice. Additionally, it was observed that *Euterpe oleracea* upregulated Nrf2 in granulosa cells, leading to an improved antioxidant capacity. These results suggest that the beneficial effects of *Euterpe oleracea* on the ovary may be attributed, at least in part, to the upregulation of the Nrf2 pathway [104].

Sulforaphane (SFN) has known antioxidant properties. It increased the expression of antioxidant enzymes (SOD, CAT), reduced ROS levels and apoptosis, and exerted a protective effect against oxidative stress in human granulosa cells by affecting the Nrf2 signaling pathway. So, SFN may have the potential to be applied in assisted reproduction cycles to enhance the quality of granulosa cells (GCs) and the oocyte [105,106].

The plant antioxidant lycopene induced an increase in the expression of Nrf2 and HO-1, promoted cellular antioxidant defenses, inhibited apoptosis, and promoted cell proliferation, thereby rescuing D-gal-induced ovarian aging [47]. Rutin, another plant antioxidant, has anti-radiation and anti-free radical effects. A study showed that rutin inhibited both natural aging-related and D-gal-induced apoptosis and ferroptosis via the Nrf2/HO-1 pathway [107]. Daphnetin is a type of traditional Chinese herbal medicine. In a concentration-dependent manner, it has the ability to safeguard ovarian function and promote follicular development in POF mice by activating the Nrf2 pathway. Moreover, the ability of daphnetin to activate the Nrf2 pathway was attenuated or even lost in Nrf2 knockout mice, suggesting that Nrf2 was the important link in the antioxidant function and rescue of POF via daphnetin [108].

Curcumin, derived from turmeric rhizomes (*Curcuma longa Linn*), possesses antioxidant, anti-inflammatory, anti-apoptotic, and antibacterial properties. It inhibited D-gal-induced oxidative stress and apoptosis in granulosa cells, leading to an increase in follicle numbers, ovarian AMH expression levels, and rescued POF in mice. This effect was achieved via the activation of the Nrf2/HO-1 and PI3K/Akt signaling pathways [109].

Puerarin, the primary bioactive compound found in puerarin isoflavone, possesses diverse pharmacological properties, including safeguarding vascular endothelium, combating aging, and displaying estrogenic activities. Puerarin increased the expression levels of SOD and Nrf2, alleviated oxidative stress, and reduced apoptosis, thereby rescuing the ovarian reserve in POF mice [110].

Pterostilbene demonstrates anti-inflammatory and cardiovascular protective properties. Pterostilbene protected human GCs from oxidative stress damage and ferroptosis by activating the Nrf2/HO-1 pathway [111].

Resveratrol is a natural phenol derived from plants that has been shown to positively influence redox flow in lipid metabolism. It reduced oogonial stem cell loss in an ovarian aging model. In vitro experiments confirmed that the activation of Nrf2 attenuated H_2_O_2_-induced cytotoxicity and oxidative stress damage [112]. Similarly, resveratrol attenuated oxidative stress and reduced apoptosis via SIRT1/Nrf2/ARE signaling in granulosa cells [113]. In short-lived fish, the administration of resveratrol resulted in the upregulation of SIRT1 and Nrf2 expressions, leading to a reduction in inflammation and endoplasmic reticulum stress. Additionally, resveratrol decreased the occurrence of ovarian follicle atresia and downregulated the expression of senescence-related markers. Consequently, the process of ovarian aging was decelerated in fish treated with resveratrol [114]. Similarly, in hens, resveratrol activated the Nrf2-ARE signaling pathway, thereby reducing tert-butyl hydroperoxide (tBHP)-induced decrease in egg production, the ovarian index, and serum E_2_ [115].

Se-enriched *Cardamine violifolia* (SEC) has excellent antioxidant properties. SEC increased the mRNA expression of Nrf2, HO-1, and NQO1, while also downregulating the mRNA expression of Keap1 in the ovaries of aging laying hens. Meanwhile, SEC had the potential to enhance egg production and egg quality. This improvement was achieved via the modulation of the Nrf2/Keap1 signaling pathway, which ultimately enhanced the antioxidant capacity of the ovary [116].

Epigallocatechin gallate (EGCG) and theaflavins (TFs) are the key molecules derived from green tea or black tea. EGCG and TFs improved ovarian endocrine function and preserved the ovarian reserve by alleviating ovarian DNA damage in CTX-exposed mice. Specifically speaking, it can activate the Nrf2/HO-1 pathways and reduce apoptosis in developing follicles [117]. Additionally, EGCG demonstrated a partial mitigation of the compromised antioxidant capacity caused via vanadium exposure, and effectively counteracted the negative impacts of vanadium on the reproductive performance and egg quality of laying hens. This beneficial effect was achieved via the activation of the Keap1/Nrf2/sMaf pathway [118].

The golden needle mushroom (*Flammulina velutipes* stem, FVS) has been reported to have anti-inflammatory, antioxidant, and hypolipidemic properties. Dietary supplementation of FVS reduced MDA levels and increased the expression of antioxidant enzymes (NQO1, GPX1, and SOD1) by activating the Keap1/Nrf2/ARE signaling pathways, leading to ameliorating the egg production rate in aging laying hens [119].

Icariin has demonstrated its involvement in the pathogenesis of several autoimmune disorders. It alleviated the injury of ovarian structure and endocrine function in autoimmune POI mice by upregulating the expression of Nrf2 and HO-1 [120].

Genistein (GEN) has received widespread attention for its multiple beneficial effects such as antioxidant and antitumor activity. Research showed that GEN enhanced the activities of antioxidant enzymes (CAT, GSH-Px, and T-SOD) and inhibited the over-accumulation of MDA by modulating the Nrf2/Keap1 pathway. Consequently, these regulatory mechanisms significantly augmented the antioxidant capacity of the ovary [59]. Daidzein, another classical isoflavone phytoestrogen, increased the protein expression of Nrf2, HO-1, and NQO1 in the ovary, thereby maintaining normal ovarian function [121].

Squid ink polysaccharide (SIP) is documented to prevent other organ dysfunctions associated with chemotherapy [122]. Liu et al. found that SIP alleviated CP-induced ovarian failure by modulating the Nrf2/ARE signaling pathway. This modulation resulted in an increase in the production of downstream factors, namely NQO-1 and HO-1, thereby enhancing the antioxidant capacity [96].

Astaxanthin supplementation has been found to decrease apoptosis and improve the antioxidant capacity of laying hens by increasing the expression of Nrf2 in the ovary. This led to an increase in serum hormone levels, ultimately slowing down the progression of ovarian aging [123].

### 4.4. Other Drugs and Treatment Modalities

Chitooligosaccharide-zinc (COS·Zn) is a powerful antioxidant and anti-aging scavenger; it modulated the SESN2/Nrf2 signaling pathway to reduce apoptosis, and improved follicle development and protected ovarian function in POF mice [124].

Glycogen synthase kinase-3 (GSK-3) inhibition can regulate oxidative stress [125]. SB216763 (a small, selective, and potent GSK-3 inhibitor) increased Nrf2 expression. This led to a decrease in the apoptotic index of the cells and a reversal of the adverse effects caused by DOX, including hormone disruption, decline in follicle number, and deterioration in oocyte quality [95].

Dimethylfumarate (DMF) is an activator of the Nrf2 pathway, and previous experiments had confirmed its ability to alleviate oxidative stress in granulosa cells. The administration of DMF to aged mice resulted in a notable increase in the quantity of follicles and the rate of follicle collection. Additionally, there was a significant rise in serum AMH levels, indicating that DMF effectively enhanced the ovarian reserve in the ovaries of aged mice. These positive effects were attributed to the activation of the Nrf2/Keap1 pathway via DMF [126].

Previous studies have shown that human placental mesenchymal stem cells (hPMSCs) can improve ovarian function in POI mice, but the mechanism by which they treat POI is unclear.

Ding et al. observed that hPMSCs exhibited a greater secretion of the cytokine epidermal growth factor (EGF) compared to other cell types. The researchers further discovered that EGF played a role in inhibiting oxidative stress by upregulating the Nrf2/HO-1 pathway. Additionally, the administration of EGF was found to enhance ovarian function in POI mice [127].

These experimental studies have shown that the Nrf2 signaling pathway mediated by these treatments against oxidative damage related mitochondrial dysfunction, apoptosis, and ferroptosis (Table 1). In chemotherapy-related ovarian aging, certain treatments have demonstrated the ability to mitigate this condition by activating the Nrf2 pathway. However, it is worth noting that Nrf2 expression is increased in various cancer cells [128,129], including pancreatic ductal carcinoma [129]. Interestingly, the depletion of Nrf2 has been found to enhance the sensitivity of tumor cells to chemotherapeutic agents [129]. Above all, the Nrf2 signaling pathway activation is a hopeful therapeutic strategy for ovarian aging. However, further investigation is still required to determine the optimal approach in chemotherapy-related ovarian aging. This is necessary in order to protect normal cells from chemotherapeutic damage, while simultaneously avoiding the induction of chemotherapeutic resistance in tumor cells.

## 5. Conclusions and Prospective Progression

Increased oxidative stress is an important hallmark of aging. Nrf2, a signaling molecule that regulates oxidative stress, exhibits an important role in the regulation of ovarian aging. As a protective molecule, Nrf2 plays a vital role in both age-related ovarian aging and pathological ovarian aging by regulating the expression of antioxidant genes. In order to offer valuable insights and suggest a promising avenue for clinical intervention in ovarian aging, we summarized a variety of natural compounds and various anti-aging therapies to mediate the activation of Nrf2. The activation of Nrf2 can effectively delay the progression of ovarian aging, protect female reproductive function, and prolong the healthy lifespan of women by reducing oxidative stress, ferroptosis, etc., thus increasing the antioxidant capacity of the ovary. However, some limitations need to be clarified. Firstly, although the aforementioned treatments do stimulate the Nrf2 pathway, the precise mechanism of Nrf2 activation in ovarian aging requires further investigation. Furthermore, more research is imperative to ascertain whether these therapeutic interventions rely on the Nrf2 pathway or alternative mechanisms are implicated. Therefore, further studies are needed to explore methods of activating Nrf2 in order to protect normal cells from damage without inducing drug resistance in tumor cells. Moreover, the current therapeutic approaches for modulating Nrf2 are primarily limited to animal experiments, and their clinical significance requires further validation.

## Figures and Tables

**Figure 1 ijms-24-13327-f001:**
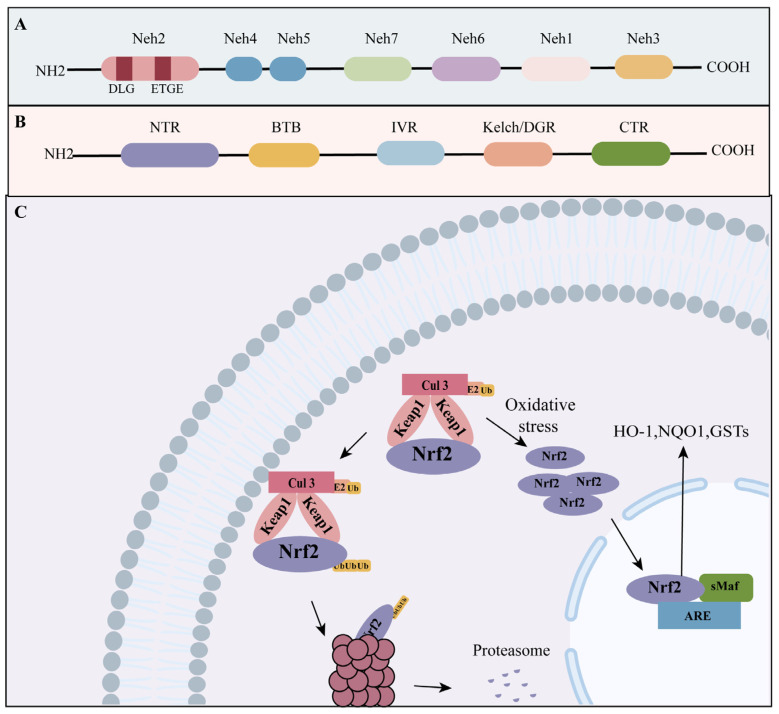
The structural architecture of Nrf2 and Keap1, and the Nrf2/Keap1 pathway. (**A**) Nrf2 consists of seven domains. (**B**) Keap1 contains five domains. (**C**) Under basal conditions, Nrf2 is bound with Keap1 and degraded in the cytosol. Under oxidative stress conditions, free Nrf2 is promoted and translocates into the nucleus and promotes the expression of downstream antioxidant genes. Some of the material in the image was sourced from BioRender.com, accessed on 24 July 2023. The black line in Figure 1C represents promotion. antioxidant response element: ARE; Kelch-like ECH-associated protein 1: Keap1; nuclear factor erythroid 2-related factor 2: Nrf2; small musculoaponeurotic fibrosarcoma: sMAF; heme oxygenase 1: HO-1; NADP (H) quinone dehydrogenase 1: NQO1; glutathione-S-transferases: GSTs.

**Figure 2 ijms-24-13327-f002:**
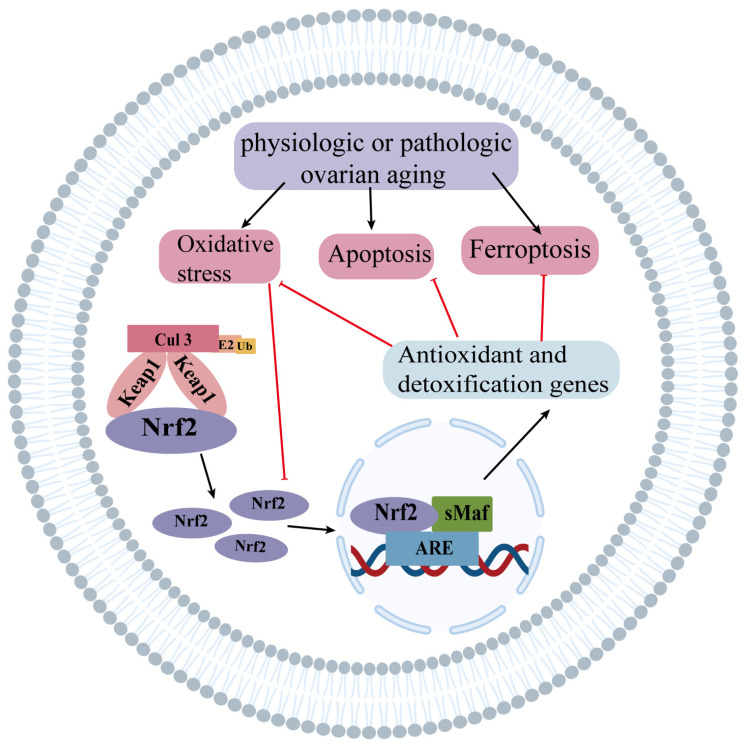
The role of the Nrf2 signaling pathway in ovarian aging. When exposed to oxidative stress, the Nrf2 pathway is active to protect against oxidative stress damage under basal conditions. However, oxidative stress increases during ovarian aging, but the Nrf2 pathway is inhibited, so apoptosis and ferroptosis increase. Some of the material in the image was sourced from BioRender.com, accessed on 20 August 2023. The red line represents inhibition and the black line represents promotion.

**Figure 3 ijms-24-13327-f003:**
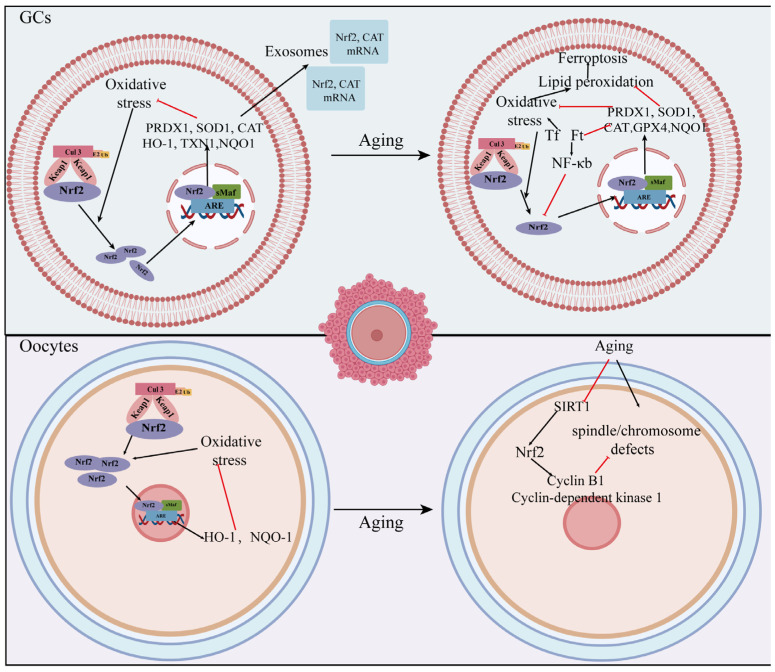
The role of the Nrf2 signaling pathway in age-related ovarian aging. When exposed to oxidative stress, the Nrf2 pathway is active to protect against oxidative stress damage. Nrf2 declines with age in both oocytes and granulosa cells, and cellular antioxidant capacity is reduced, which influences cellular oxidative stress, apoptosis, and ferroptosis, eventually affecting cellular function. Some of the material in the image was sourced from BioRender.com, accessed on 24 July 2023. The red line represents inhibition and the black line represents promotion. peroxiredoxin 1: PRDX1; superoxide dismutase1: SOD1; catalase: CAT; heme oxygenase 1: HO-1; thioredoxin 1: TXN1; NADP (H) quinone dehydrogenase 1: NQO1; glutathione peroxidase 4: GPX4.

**Figure 4 ijms-24-13327-f004:**
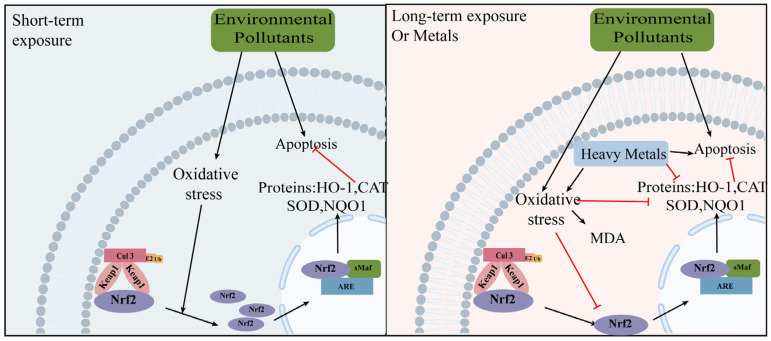
The role of the Nrf2 signaling pathway in environmental pollutant-related ovarian aging. When short-term exposed to environmental pollutants, the Nrf2-Keap1 pathway is activated to defend against oxidative damage in the early stage of oxidative stress, but when long-term exposure occurs, the antioxidant system cannot eliminate the excess reactive oxygen species in the cell, and it will in turn reduce the activity of antioxidant enzymes, inhibit the Nrf2/Keap1 pathway, and exacerbate ovarian toxicity. Heavy metals may directly reduce antioxidant enzyme activity to damage cells. Some of the material in the image was sourced from BioRender.com, accessed on 24 July 2023. The red line represents inhibition and the black line represents promotion. superoxide dismutase1: SOD1; catalase: CAT; heme oxygenase 1: HO-1; NADP (H) quinone dehydrogenase 1: NQO1; malondialdehyde: MDA.

**Figure 5 ijms-24-13327-f005:**
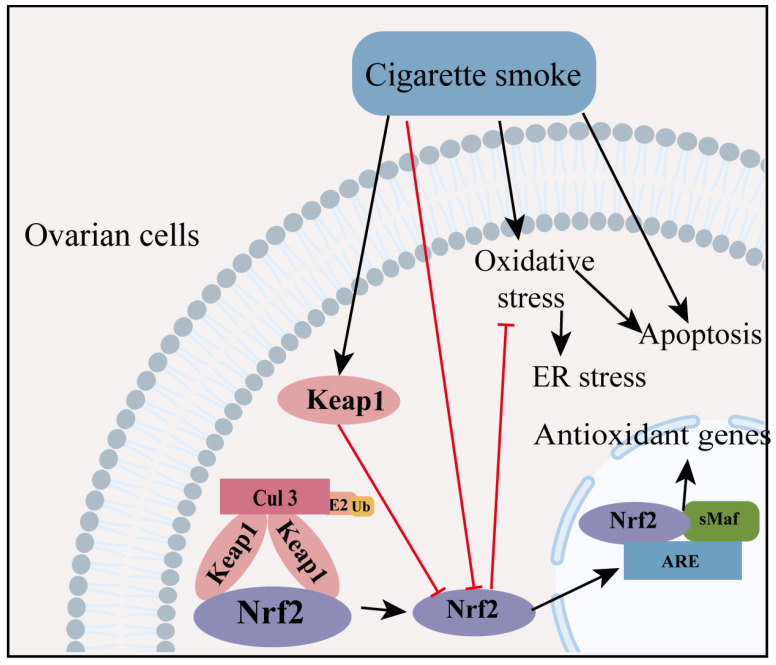
The role of the Nrf2 signaling pathway in unhealthy lifestyle-related ovarian aging. CSE induces a cytostatic effect regulated by cell-cycle associated genes, and leads to oxidative stress via the Keap1/Nrf2 pathway, eventually resulting in the death of ovarian cells. Some of the material in the image was sourced from BioRender.com, accessed on 24 July 2023. The red line represents inhibition and the black line represents promotion. endoplasmic reticulum: ER.

**Figure 6 ijms-24-13327-f006:**
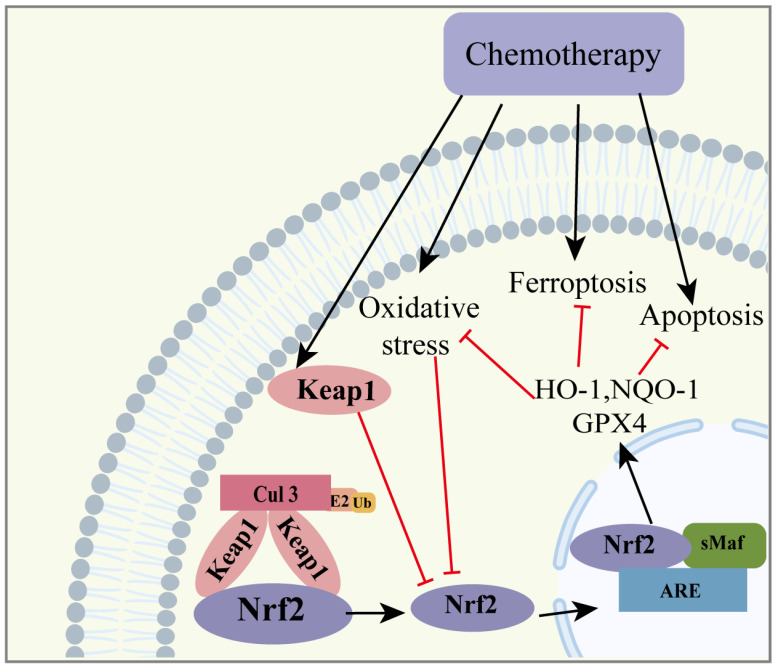
The role of the Nrf2 signaling pathway in chemotherapy-related ovarian aging. At the starting exposure to chemotherapeutic drugs, cells undergo an adaptive response that activates Nrf2 to produce antioxidant products to protect the cells. However, when the dose of chemotherapeutic drugs is increased or the exposure time is prolonged, the cellular defense response is unable to completely resist the toxicity of chemotherapeutic drugs, which eventually leads to cellular dysfunction. Some of the material in the image was sourced from BioRender.com, accessed on 24 July 2023. The red line represents inhibition and the black line represents promotion. heme oxygenase 1: HO-1; NADP (H) quinone dehydrogenase 1: NQO1; glutathione peroxidase 4: GPX4.

**Table 1 ijms-24-13327-t001:** Potential treatment for ovarian aging by targeting Nrf2.

Treatment		Models	Targets	Potential Effects
Traditional Chinese Medicinal Treatment	Electroacupuncture [97]	Rats	Keap1/Nrf2/HO-1 pathway	Electroacupuncture rescued CTX-induced ovarian dysfunction and reduced oxidative stress and inflammation.
	Bu Shen Huo Xue Tang [98]	Mice	Nrf2/Keap1 pathway	Bu Shen Huo Xue Tang increased the levels of E2 and AMH and decreased the levels of FSH and LH.
	Si-Wu-Tang [99]	Mice	Nrf2/HO-1 pathway	Si-Wu-Tang improved estrogen levels, follicle number, and produces better fertility outcomes in POF mice.
Nutrients or Endogenous Hormone Supplements	Vitamin C [63]	OVCAR-3	Nrf2/HO-1 pathway	Vitamin C attenuated 1-BP-induced apoptosis and inhibited caspase activity.
	Vitamin E [100]	Granulosa cells	Nrf2 pathway	Vitamin E alleviated oxidative stress-induced apoptosis and decreased intracellular ROS levels.
	Arginine [102]	Multiparous Hu sheep	Nrf2/Keap1 pathway	Arginine ameliorated the prolongation of estrous cycle days, hormone level disorders, and increased the antioxidant capacity of ovarian tissues in multiparous Hu sheep.
	N-acetylcysteine [94]	Mice	Nrf2/HO-1 pathway	N-acetylcysteine rescued cisplatin-induced ovarian aging and attenuated apoptosis and ferroptosis in GCs.
		SVOG and KGN		
	Melatonin [60]	Golden hamsters	SIRT1/Nrf2/NF-κB pathway	Melatonin ameliorated the decline in ovarian function caused by BPS, stimulated the expression of key redox/survival markers, and enhanced ovarian antioxidant capacity.
Natural Plant or Animal Extracts	*Euterpe oleracea* [104]	Mice	Nrf2 pathway	*Euterpe oleracea* improved blastocyst development ability and mitochondrial function in aged mice.
	Sulforaphane [106]	Granulosa cells	Nrf2 pathway	Sulforaphane exerted a protective effect against oxidative stress in granulosa cells.
	Lycopene [47]	Hyline-Brown laying hens	Nrf2/HO-1 pathway	Lycopene exerted antioxidant capacity, ameliorated oxidative stress, promoted cell proliferation, and inhibited D-gal-induced apoptosis in aging hen ovaries.
	Rutin [107]	Hyline-Brown laying hens	Nrf2/HO-1 pathway	Rutin rescuing age-related and D-gal-induced ovarian aging, promoted cellular antioxidant defenses, inhibited apoptosis, and promoted cell proliferation.
	Daphnetin [108]	Mice and *Nrf2*^−/−^ mice	Nrf2 pathway	Daphnetin rescued POF mice.
	Curcumin [109]	Mice	Nrf2/HO-1 pathway	Curcumin increased follicle number and inhibited d-gal-induced oxidative stress and apoptosis in granulosa cells.
	Puerarin [110]	Mice	Nrf2 pathway	Puerarin rescued ovarian reserve in POF mice, alleviated oxidative stress, and reduced apoptosis.
	Pterostilbene [111]	COV434 and KGN	Nrf2/HO-1 pathway	Pterostilbene protected human GCs from oxidative stress damage and ferroptosis.
	Resveratrol [112,113,114,115]	Mice	Nrf2 pathway	Resveratrol reduced oogonial stem cells loss in an ovarian aging model and attenuated H2O2-induced cytotoxicity and oxidative stress damage.
		Granulosa cells	SIRT1/Nrf2/ARE pathway	Resveratrol attenuated oxidative stress and reduced apoptosis in granulosa cells.
		Fish	SIRT-1/Nrf2 pathway	Resveratrol slowed down ovarian aging and reduced inflammation and ER stress in short-lived fish.
		Lohmann laying hens	Keap1/Nrf2/ARE pathway	Resveratrol reduced a tBHP-induced decrease in egg production, ovarian index, and serum E2.
	*Cardamine violifolia* [116]	Roman laying hens	Nrf2/Keap1 pathway	*Cardamine violifolia* improved egg production and egg quality in the ovaries of aging laying hens.
	Epigallocatechin gallate (EGCG) and theaflavins (TFs) [117]	Mice	Nrf2/HO-1 pathway	EGCG and TFs alleviated ovarian DNA damage in CTX-exposed mice, improved ovarian endocrine function, and rescued the decrease in ovarian reserve.
	Epigallocatechin gallate [118]	Hyline-Brown laying hens	Keap1/Nrf2 pathway	EGCG partially alleviated the vanadium-induced decrease in antioxidant capacity and reversed the adverse effects of vanadium on laying performance and egg quality in laying hens’ ovaries.
	*Flammulina velutipes* [119]	Hyline-Brown laying hens	Keap1/Nrf2/ARE pathway	Dietary supplementation of *Flammulina velutipes* ameliorated the egg production rate.
	Icariin [120]	Mice	Nrf2/HO-1/SIRT1 pathway	Icariin alleviated the ovarian structure and ovarian endocrine function in autoimmune POI mice.
	Genistein [59]	Hy-Line Brown laying hens	Nrf2/Keap1 pathway	Genistein promoted the growth of dominant follicles.
	Daidzein [121]	Sows	Nrf2/HO-1 pathway	Daidzein maintained normal ovarian function.
	Squid ink polysaccharide (SIP) [96]	Mice	Nrf2/ARE pathway	SIP alleviated CP-induced ovarian failure.
	Astaxanthin [123]	Hy-line brown laying hens	Nrf2 pathway	Astaxanthin ameliorated ovarian aging and improved the antioxidant capacity of laying hens.
Other Drugs and Treatment Modalities	Chitooligosaccharide-zinc [124]	Mice	SESN2/Nrf2 pathway	Chitooligosaccharide-zinc improved follicle development and protected ovarian function in POF mice.
	Glycogen synthase kinase-3 inhibitor [95]	Mice	Nrf2 pathway	Glycogen synthase kinase-3 inhibitor reversed the hormone disruption, follicle number decline, and oocyte quality decline induced by DOX.
	Dimethylfumarate [126]	Mice	Nrf2/Keap1 pathway	Dimethylfumarate improved the ovarian reserve in the ovaries of old mice.
	Human placental mesenchymal stem cells [127]	Mice	Nrf2/HO-1 pathway	Human placental mesenchymal stem cells exhibited a greater secretion of the cytokine epidermal growth factor (EGF) to improve ovarian function in POI mice.

Anti-Mullerian hormone: AMH; antioxidant response element: ARE; Bisphenol S: BPS; cyclophosphamide: CP; cyclophosphamide: CTX; doxorubicin: DOX; epigallocatechin gallate: EGCG; epidermal growth factor: EGF; endoplasmic reticulum: ER; estrogen: E_2_; follicle stimulating hormone: FSH; granulosa cells: GCs; heme oxygenase 1: HO-1; Kelch-like ECH-associated protein 1: Keap1; luteinizing hormone: LH; nuclear factor kappa-light chain-enhancer of activated B lymphocytes: NF-κB; nuclear factor erythroid 2-related factor 2: Nrf2; premature ovarian failure: POF; premature ovarian insufficiency: POI; reactive oxygen species: ROS; sestrin 2: SESN2; Squid ink polysaccharide: SIP; sirtuin 1: SIRT1; theaflavins: TFs; tert-butyl hydroperoxide: tBHP; 1-Bromopropane: 1-BP.

## Data Availability

Not applicable.

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
