# Peer review of "Role of the Nrf2 Signaling Pathway in Ovarian Aging: Potential Mechanism and Protective Strategies"

_ijms, 2023, doi:10.3390/ijms241713327_

Round 1

Reviewer 1 Report

The manuscript is interesting, generally well written and well illustrated but some improvements are necessary. In particular:

Introduction: Since this is a review article, the multifaceted role of NRF2/KEAP1 signalling deserves to be highlithed. In fact, it deserves to be pointed out that NRF2/KEAP1 signalling pathway play a key role in several cancerous and non-cancerous diseases (as recently reviewed PMID: 37296665, 35453348 , 36289931 , 37525922 , 37519172 ). 

4. Targeting Nrf2 signaling as therapeutic interventions in ovarian aging: a summary table reporting the main results of the studies discussed should be added. 

A schematic representation of Nrf2 signaling pathway in ovarian aging and its modulation should be added

Author Response

Response to Reviewer 1 Comments

Thank you for your thoughtful and constructive reviews. In response to your comments, we have added new table, revised the text, and modified the figures, as detailed in our responses below. All major changes in the text are highlighted with yellow. We have tried our best to address all of the comments, and we believe that the manuscript has improved in the process.

Point 1: Introduction: Since this is a review article, the multifaceted role of NRF2/KEAP1 signalling deserves to be highlithed. In fact, it deserves to be pointed out that NRF2/KEAP1 signalling pathway play a key role in several cancerous and non-cancerous diseases (as recently reviewed PMID: 37296665, 35453348 , 36289931 , 37525922 , 37519172 ). 

Response 1: We gratefully appreciate your valuable suggestion! According your suggestion, we have added a small summary of the role of Nrf2 in other diseases. For example, “As an important regulator of oxidative stress, the Nrf2/Keap1 pathway also plays an important role in aging and various diseases, including preeclampsia, traumatic brain injury, and cancer. In aging and some diseases, the Nrf2/Keap1 pathway is a protective factor, but may also act as a double agent in cancer.” Thanks for the references you offered, and after literature reading and searching, we have added more informations to the manuscript highlighted with yellow (page 2, line 62-66).

Point 2: 4. Targeting Nrf2 signaling as therapeutic interventions in ovarian aging: a summary table reporting the main results of the studies discussed should be added.

Response 2: Thank you for your sincere suggestion. We agree with your opinion that a summary table can present the content of the review better. We have summarized the therapeutic approaches targeting Nrf2 and added a table in Part 4. (page 14-16, line 520-529)

Point 3: A schematic representation of Nrf2 signaling pathway in ovarian aging and its modulation should be added

Response 3: Thank you for your sincere advice. We have summarized the role of Nrf2 in physiologic and pathologic ovarian aging and drawn a schematic diagram (page 4, line 119-122).

We tried our best to improve the manuscript and made some changes in the manuscript. These changes will not influence the content and framework of the paper. We appreciate for Editors’ and Reviewers’ warm work earnestly, and hope that the correction will meet with approval. Once again, thank you very much for your comments and suggestions.

Reviewer 2 Report

The manuscript “Role of Nrf2 signaling pathway in ovarian aging: potential Mechanism and protective strategies” by Gao et al. provides a comprehensive account on the adverse effects of oxidative stress on ovarian health and ageing. The review also discusses in great detail the role of Nrf2/Keap1 pathway in ovarian ageing as well as various therapeutic approaches modulating the Nrf2 pathway to alleviate ovarian aging.

  1. Title- Mechanism- “M” should not be capital.

  2. Abstract- Change this sentence “Ovary is an important reproductive endocrine organ of women, and its aging will lead to menopause, decreased fertility, and long-term hazards such as osteoporosis, cardiovascular diseases and cognitive impairment. Oxidative stress is associated with aging metabolism” to “The ovary holds a significant role as a reproductive endocrine organ in women, and its aging process bears implications such as menopause, decreased fertility, and long-term health risks including osteoporosis, cardiovascular disorders, and cognitive decline. The phenomenon of oxidative stress is tightly linked to the aging metabolic processes.”

  3. Abstract section- “Notably, the age-related function decline of ovary occurs earlier than other organs” Authors could mention here why is this the case. For example, Unlike other organs, which can undergo tissue regeneration, the ovary has a limited pool of ovarian follicles at birth. As these follicles are used up over time, there's no significant regenerative capacity for the ovary to counteract the effects of aging. It will help highlight the additional burden of aging on ovaries as an organ.

  4. Various sentences throughout the manuscript are quite long and dense. Break down some of these complex sentences into smaller ones for better readability and clarity. This will help the reader absorb the information more easily.

  5. The manuscript lacks flow between ideas at multiple places. Try to use transition sentences or phrases. For instance, between the discussion of ovarian aging and the transition to oxidative stress, authors can discuss the effect of oxidative stress on ovarian aging a bit more summarizing the connection between the two topics.

  6. Provide references- “Many studies have confirmed that Nrf2 acts as a protective factor against physiological and pathological ovarian aging and as a cellular defense against the effects of aging”.

  7. Is this in context to ovarian environment- “Nrf2 is an important regulator of cellular antioxidant responses and its expression showed a decreasing trend with increasing age [35-37].”

  8. “inflamm-aging” introduce the term before directly using it in the manuscript.

  1. In women, they even reduce ovarian reserve function, and mediate the onset of ovarian aging [45,46]- Authors can add more such studies here showing the adverse effects of wide range of xenobiotics on the ovarian health, for example, PMID: 30877919.

  2. Line 220 “But it comes an increased concerns about their biosafety” reframe the sentence. Also discuss other concerns about the use of nanoparticles in the context of this manuscript.

  3. Lines 232-242- This whole section summarizes the role of Nrf2/Keap1 pathway in ovarian ageing against various environmental pollutants therefore it will be better to write it as another section rather than under nanoparticles.

  4. “The number of female smokers is gradually rising, but smoking is not good for women's health, not only increases the risk of lung cancer, but also damage to women's reproductive capacity[66].” Reframe the sentence, Smoking is not good for anyone not just females. Authors may say that the percentage of women among the total smoking population is on rise and as harmful effects of smoking are well established it has major concerns for females due to its adverse effects on reproductive system.

  5. Line 330- “Bu Shen Huo Xue Tang” introduce what Bu Shen Huo Xue Tang is and how it has been used in traditional Chinese medicine (TCM) clinically to treat POI.

There are couple of minor typos, spelling and grammar mistakes: Line 269- Further spelling mistake, Line 306- "So, chemotherapeutic drugs" should be "Therefore, chemotherapeutic drugs". 

Author Response

Response to Reviewer 2 Comments

Thank you for your thoughtful and constructive reviews. In response to your comments, we have revised the text, and modified the figures, as detailed in our responses below. All major changes in the text are highlighted with green. We have tried our best to address all of the comments, and we believe that the manuscript has improved in the process.

Point 1: Title- Mechanism- “M” should not be capital.

Response 1: Thank you for this kind remind! We are very sorry for our incorrect writing. We have carefully corrected the minor typos, spelling and grammar mistakes in the manuscript. After checking the revised manuscript again and again, we couldn’t find spelling errors.

Point 2: Abstract- Change this sentence “Ovary is an important reproductive endocrine organ of women, and its aging will lead to menopause, decreased fertility, and long-term hazards such as osteoporosis, cardiovascular diseases and cognitive impairment. Oxidative stress is associated with aging metabolism” to “The ovary holds a significant role as a reproductive endocrine organ in women, and its aging process bears implications such as menopause, decreased fertility, and long-term health risks including osteoporosis, cardiovascular disorders, and cognitive decline. The phenomenon of oxidative stress is tightly linked to the aging metabolic processes.”

Response 2: Thank you! It is a nice advice. We have made revisions to our formulation in order to enhance the clarity and precision of our intended meaning (page 1, line 10-13). Thanks again for your kind advice.

Point 3: Abstract section- “Notably, the age-related function decline of ovary occurs earlier than other organs” Authors could mention here why is this the case. For example, Unlike other organs, which can undergo tissue regeneration, the ovary has a limited pool of ovarian follicles at birth. As these follicles are used up over time, there's no significant regenerative capacity for the ovary to counteract the effects of aging. It will help highlight the additional burden of aging on ovaries as an organ.

Response 3: Thank you for this nice suggestion! We agree with your opinion that this statement is lack of details. In the revised manuscript, we have adjusted this sentence to “Unlike other organs, which can undergo tissue regeneration, the ovary has a limited pool of ovarian follicles at birth. As these follicles are used up over time, there's no significant regenerative capacity for the ovary to counteract the effects of aging. Studies have found that reproductive aging occurs prior to somatic aging” (page 1, line 25-29). Thanks again for your kind advice.

Point 4: Various sentences throughout the manuscript are quite long and dense. Break down some of these complex sentences into smaller ones for better readability and clarity. This will help the reader absorb the information more easily.

Response 4: Thank you for your kind reminds! We have fully checked the manuscript again and made revisions to the long and complex sentences, such as page 6 line 190-194, page 6 line 212-215, page 8 line 288-294.

Point 5: The manuscript lacks flow between ideas at multiple places. Try to use transition sentences or phrases. For instance, between the discussion of ovarian aging and the transition to oxidative stress, authors can discuss the effect of oxidative stress on ovarian aging a bit more summarizing the connection between the two topics.

Response 5: Thank you for your sincere suggestion. Based on your suggestions, we have included some linking sentences or phrases to make the ideas flow better. For example, a brief discussion precedes the introduction of oxidative stress. For example, “Numerous researches have demonstrated that the ovarian microenvironment's oxidative stress condition can lead to pathological harm. The harm includes the in-terruption of meiosis in oocytes, apoptosis of granulosa cells, and dysfunction of the corpus luteum. Consequently, this expedites the progression of ovarian aging” (page 2, line 46-49).

Point 6: Provide references- “Many studies have confirmed that Nrf2 acts as a protective factor against physiological and pathological ovarian aging and as a cellular defense against the effects of aging”.

Response 6: Thank you for your sincere reminds. Now we have inserted the relevant references(PMID: 36050696, 26247513, 29117219, 30368232)(page 3, line 113-115). Thanks again for your kind advice.

Point 7: Is this in context to ovarian environment- “Nrf2 is an important regulator of cellular antioxidant responses and its expression showed a decreasing trend with increasing age [35-37].”

Response 7: We gratefully appreciate for your kind reminds! Three studies experimentally confirmed decreased Nrf2 expression in ovarian tissue of aged animals. We have changed the formulation of the original sentence to present our idea more clearly. As follows, “Nrf2 is an important regulator of cellular antioxidant responses, and its expression showed a decreasing trend with increasing age in ovarian tissues.”(page 4, line 129-130).

Point 8: “inflamm-aging” introduce the term before directly using it in the manuscript.

Response 8: Thank you for your sincere suggestion. Inflammatory aging (inflamm-aging) is a persistent and mild inflammatory condition that develops as age increases. In recent years, research has indicated that the process of inflammatory aging is a significant factor in the development of premature ovarian insufficiency. We have already given a brief introduction to this concept before the term was first used in the revised manuscript (page 5, line 143-146).

Point 9: In women, they even reduce ovarian reserve function, and mediate the onset of ovarian aging [45,46]- Authors can add more such studies here showing the adverse effects of wide range of xenobiotics on the ovarian health, for example, PMID: 30877919.

Response 9: Thank you for your sincere advice. As you can see, the literature we previously used to confirm that environmental pollutants induced ovarian aging was not enough. Thanks for the references you offered, and we have added more references to the manuscript (page 6, line 180-181).

Point 10: Line 220 “But it comes an increased concerns about their biosafety” reframe the sentence. Also discuss other concerns about the use of nanoparticles in the context of this manuscript.

Response 10: Thank you for your sincere suggestion. Firstly, we have re-written this sentence according to your suggestion, as follows, “Therefore, more attention should be given to biosecurity of nanoparticles.” Lastly, we have added the discussion of concerns about the use of nanoparticles in the revised manuscript, for example, “As nanoparticles become more prevalent in the environment, the likelihood of exposure to nanoparticles increases. Recently, nanoparticles have been found to cause harmful effects when ingested by humans or animals. They cause damage to the digestive, respiratory, and nervous systems” (page 7, line 244-248).

Point 11: Lines 232-242- This whole section summarizes the role of Nrf2/Keap1 pathway in ovarian ageing against various environmental pollutants therefore it will be better to write it as another section rather than under nanoparticles.

Response 11: Thank you for this good suggestion. It is really true as you suggested that the summary section as another section is a better choice. Therefore, we have wrote this summary section as “3.2.5 Conclusion”( page 7, line 260).

Point 12: “The number of female smokers is gradually rising, but smoking is not good for women's health, not only increases the risk of lung cancer, but also damage to women's reproductive capacity[66].” Reframe the sentence, Smoking is not good for anyone not just females. Authors may say that the percentage of women among the total smoking population is on rise and as harmful effects of smoking are well established it has major concerns for females due to its adverse effects on reproductive system.

Response 12: Thank you for bringing this important issue to our attention.We have revised the text to address your concerns and hope that it is now clearer. We have changed original sentences to “The percentage of women among the total smoking population is on rise. As the harmful effects of smoking are well established, it has major concerns for females due to its adverse effects on the reproductive system” ( page 8, line 283-285).

Point 13: Line 330- “Bu Shen Huo Xue Tang” introduce what Bu Shen Huo Xue Tang is and how it has been used in traditional Chinese medicine (TCM) clinically to treat POI.

Response 13: Thank you for your kind advice. We have added the information about Bu Shen Huo Xue Tang in the manuscript. For example, “The traditional Chinese medicine formula Bu Shen Huo Xue Tang (BSHXT) is a modi-fied version of the traditional Qing'e Pill formula from the Song Dynasty's "Taiping Huimin and Agent Bureau". What's more, it is commonly employed in clinical settings for the management of POI” (page 11, line 360-363).

We tried our best to improve the manuscript and made some changes in the manuscript. These changes will not influence the content and framework of the paper. We appreciate for Editors’ and Reviewers’ warm work earnestly, and hope that the correction will meet with approval. Once again, thank you very much for your comments and suggestions.